# Acute Effects of Different Activity Types and Work-To-Rest Ratio on Post-Activation Performance Enhancement in Young Male and Female Taekwondo Athletes

**DOI:** 10.3390/ijerph19031764

**Published:** 2022-02-04

**Authors:** Ibrahim Ouergui, Slaheddine Delleli, Hamdi Messaoudi, Hamdi Chtourou, Anissa Bouassida, Ezdine Bouhlel, Emerson Franchini, Luca Paolo Ardigò

**Affiliations:** 1High Institute of Sport and Physical Education of Kef, University of Jendouba, El Kef 7100, Tunisia; ouergui.brahim@yahoo.fr (I.O.); bouassida_anissa@yahoo.fr (A.B.); 2Institut Supérieur du Sport et de l’Education Physique de Sfax, Université de Sfax, Sfax 3000, Tunisia; sdelleli2018@gmail.com (S.D.); hamdimessaoudihamdi@gmail.com (H.M.); h_chtourou@yahoo.fr (H.C.); 3Activité Physique, Sport et Santé, UR18JS01, Observatoire National du Sport, Tunis 1003, Tunisia; 4Laboratoire de Physiologie de l’exercice et Physiopathologie, de L’intégré au Moléculaire “Biologie, Médecine, Santé”, UR12ES06, Faculty of Medicine Ibn El Jazzar, University of Sousse, Sousse 4000, Tunisia; ezdine_sport@yahoo.fr; 5Martial Arts and Combat Sports Research Group, School of Physical Education and Sport, University of São Paulo, 05508-030 São Paulo, Brazil; efranchini@usp.br; 6Department of Neurosciences, Biomedicine and Movement Sciences, School of Exercise and Sport Science, University of Verona, Via Felice Casorati 43, 37131 Verona, Italy

**Keywords:** repeated technique, plyometrics, combat sports, agility, PAPE

## Abstract

The study assessed conditioning activities’ (CAs’) effects involving different work-to-rest ratios (WRR) on taekwondo athletes’ physical performance. Adolescent taekwondo athletes (age: 16 ± 1 years) randomly participated in the control six experimental conditions. Each condition was composed of standard warm-up and CA composed of plyometrics (P) or repeated high-intensity techniques (RHIT) performed using three different WRR: 1:6, 1:7, self-selected rest time (SSRT). After rest, athletes performed countermovement jump (CMJ), taekwondo specific agility test (TSAT), 10 s frequency speed kick test (FSKT-10s), multiple frequency speed kick test (FSKT-mult). P1:7, SSRT induced techniques higher number in FSKT-10s (*p* < 0.001 for all comparisons) and lower TSAT time (*p* < 0.01 for all comparisons) compared with control. Kicks-number recorded during FSKT-mult was lower in the control compared with RHIT1:6 (*p* = 0.001), RHIT1:7 (*p* < 0.001), RHITSSRT (*p* < 0.05), P1:7 (*p* < 0.001), and SSRT (*p* < 0.001). Kicking decrement index (DI) during FSKT-mult was lower after RHIT1:6 compared with control (*p* = 0.008), RHIT1:7 (*p* = 0.031), P 1:6 (*p* = 0.014), PSSRT (*p* = 0.041). (1) P1:7 and PSSRT can be used to improve taekwondo-specific agility and kicks-number, (2) RHIT1:6 is beneficial to maintain low DI, and (3) plyometric and different WRR-repeated-techniques can enhance kicks-number.

## 1. Introduction

Taekwondo is an intermittent striking combat sport requiring kick techniques performed with high intensity, jumps, and changes of direction to control distance from the opponent during the preparation to deliver subsequent kicks [1]. During the match, these high-intensity actions are interspersed by low-intensity actions or pauses [2], resulting in a work-to-rest ratio varying from 1:2 to 1:8 [2,3]. Regarding physical aspects, scoring actions rely mainly on muscle power [1]. Therefore, several strategies have been proposed to optimize muscle power; among them, post-activation potentiation (PAP) and post-activation performance enhancement (PAPE) have been suggested as two different effective methods to induce performance improvements [4,5]. The term PAP can be used to indicate the short-term gains in muscular force/torque production during an electrically evoked twitch [4], which is largely attributed to myosin light chain phosphorylation within type II fibers [6]. The increase in twitch torque lasts only few minutes (<3 min) after the conditioning activity and declines rapidly [6]. However, the peak voluntary performance improvement is often observed between 7 and 10 min after conditioning activity in most studies [7]. For this reason, actual investigations [4,5] revealed the confusion between the PAP and the PAPE, where the enhancements in voluntary muscular force production results from muscle temperature changes, intramuscular fluid accumulation, motor learning, and the increase in spinal-level excitability (motoneuron output).

In combat sports, conditioning activity with different procedures has been used in judo [8,9], karate [10] Muay Thai [11], and taekwondo [12,13,14,15]. In this context, plyometrics (i.e., exercises during which muscles develop maximum force over a short time to increase power (force·speed)) have been widely proposed as a conditioning activity, which induces significant improvement in subsequent performance [8,9,10]. It has been shown that plyometrics-based conditioning activity improves jump height by 3.5% in karate athletes [10] and increases the number of throws by 12% in the Special Judo Fitness Test [9]. Additionally, performing repeated techniques was reported to be a good stressor for the neuromuscular system [11], showing a potential to enhance performance [11]. Involving combat sport kicks, Aandahl et al. [12] showed a positive effect of kicks resisted by elastic bands on athletes’ performance. Whereas specific techniques were used to prepare athletes to handle the physical and physiological strains of their sport [16], this type of activity has not yet been well introduced as a conditioning activity during warm-up sessions to induce potentiation in combat sports. Moreover, potentiation is the result of the balance between the mechanisms of PAPE and fatigue [6,7]. Whereas numerous studies [13,14,15,17,18] have investigated the effect of different factors that can codify this balance (e.g., volume, intensity and rest interval), the optimal work-to-rest ratios within each conditioning activity have not yet been determined in the scientific literature, especially regarding performance specific to combat sports. Due to the intermittent nature of taekwondo, alternating effort and rest times using specific ratios can be a very important factor that can affect subsequent specific performances [19].

Both taekwondo professionals and sport scientists aim to prepare athletes to better handle the competition demands. To do this, manipulation of different conditioning activities and work-to-rest ratios can be pursued to administer more effective training. To the current authors’ knowledge, no studies have investigated the combined effects of using plyometrics (P) or repeated high-intensity techniques (RHIT) with different work-to-rest ratios during warm-up sessions on subsequent specific taekwondo performances. For this reason, the present study aims to compare the effects of six different conditioning activities (P vs. RHIT) using different work-to-rest ratios (1:6, 1:7, and self-selected rest time, SSRT) and a control condition on physical performances in young taekwondo athletes. Due to the beneficial effects of the specific technical work on the technical performance of taekwondo athletes [16], it was hypothesized that RHIT with a specific ratio (1:6 or 1:7) would potentiate performance in 10 s frequency of kicks test (FSKT-10s) and in its multiple version (5 × 10 s with 10 s intervals, FSKT-mult). However, as plyometric exercise was shown to be more appropriate to improve jump ability [10] and agility performance [20], it was hypothesized that plyometric-based conditioning activity would result in better jump height and agility performances with all ratios used.

## 2. Materials and Methods

### 2.1. Participants

Following convenience sampling, twenty-seven taekwondo athletes (14 males and 13 females; mean ± SD, age: 16 ± 1 years; height: 1.69 ± 0.09 m; body mass: 60.1 ± 10.7 kg; taekwondo experience: 7 ± 1 years) belonging to the same club volunteered to participate in this study. To participate in the study, all athletes had to meet the following inclusion criteria: (i) they should be young taekwondo athletes; (ii) they should have five years or more of taekwondo experience; (iii) they should present no history of disease, medication, or lower limb injuries or neuromuscular disorders; (iv) athletes should not be in a period of body mass reduction process; and (v) females could participate if they were in the follicular phase of their menstrual cycle during the experimentation. All participants regularly participated in competitions (regional level and national) for more than 2 years and trained 2 to 3 days per week (2 h per session). Athletes did not present any medical restrictions during the experimental period and refrained from any strenuous exercises 48 h before the beginning of the experimental sessions. Participants were given an in-depth explanation of the protocol before starting the study. After explaining the advantages and disadvantages of the study, parental consent was granted, and written informed consent was signed. The study was carried out in accordance with the Declarations of Helsinki and approved by a local research ethics committee (CPP SUD N° 0332/2021).

### 2.2. Procedures

This study used a randomized repeated-measures research design to investigate the effect of conditioning activities and work-to-rest ratios on countermovement jump test (CMJ), FSKT-10s, FSKT-mult, and taekwondo-specific agility test (TSAT) performances. Due to the study design, allocation concealment as well as blinding could not be executed, as the athletes were aware of the condition being performed.

One week before the beginning of the experimentation, athletes were well familiarized with the experimental procedures. In order to overcome time-of-day effects, all testing sessions were conducted at the same time of day (5 p.m. to 7 p.m.). In a randomized fashion, taekwondo athletes were assessed using CMJ [21], FSKT-10s [15], FSKT-mult [14], and TSAT [22] after a control condition (i.e., 10 min running at 9 km/h followed by 2 min rest [15], and 10 min after each of the 6 experimental conditions. Regarding experimental conditions, after being submitted to a standard warm-up (i.e., control condition), athletes performed, to the best of their abilities, repeated high-intensity techniques (RHIT, i.e., 3 sets of alternative kicks—bandal-chagui—with a maximum of kicks possible over 5 s) or plyometric exercise (P, i.e., 3 sets of consecutive vertical jumps (both lower limbs together) as fast as possible over obstacle of 40 cm for 5 s) using different work-to-rest ratios within each conditioning activity (i.e., 1:6, 1:7 and SSRT; Figure 1).

The effort time (i.e., ~5 s) was chosen in an attempt to reflect the mean time of attacks and counter-attacks reported in taekwondo competition, and imposed ratios (i.e., 1:6 and 1:7) were chosen as they represent the time-structure of taekwondo combat [1]. All conditions were performed in separate sessions with a minimum rest interval of 24 h to avoid the effect of fatigue of previous interventions.

#### 2.2.1. Countermovement Jump Test

For the evaluation of lower-limb jumping performance, the CMJ test was executed using an infrared jump system (Optojump, Microgate, Bolzano, Italy). Athletes started from a standing position and performed a rapid downward motion by flexing the knees and hips immediately followed by a rapid extension of these joints. To exclude the influence of arm swing, subjects were asked to keep their hands on their waists. No lower limb flexion or arm swing in the upward phase was allowed. Three trials were carried out, and the best performance was maintained for analysis. The intra-class correlation coefficient (ICC) for test–retest trial for the present study was 0.97.

#### 2.2.2. Ten Seconds Frequency Speed of Kick Test

During the FSKT-10s test, each athlete must perform the maximum number of kicks against a bag by alternating the right and left lower limb. The technique used during the 10 s of test was the bandal-chagui. The total number of kicks represented the performance in this test was 15. The ICC for test–retest trial for the present study was 0.87.

#### 2.2.3. Multiple Frequency Speed of Kick Test

Each athlete performed five sets of FSKT-10s with a 10 s rest interval between repetitions [23]. Performance was determined as the total number of kicks performed in each set and the total number of kicks in 5 sets, which allowed determining the Kick Decrement Index (DI; [23]) Kick Decrement Index was calculated according to Da Silva Santos et al. [14] as follows (Equation (1)):DI (%) = [1 − ((FSKT1 + FSKT2 + FSKT3 + FSKT4 + FSKT5)/(Best FSKT set × Numbers of sets))] × 100(1)

#### 2.2.4. Taekwondo-Specific Agility Test

The athlete began the test in his/her fighting stance position and behind the start line. At his/her discretion, the athlete advanced to the center mark as fast as possible. Then, following his own choice, he turned towards partner 1 by a shift and performed a roundhouse kick with his lead leg. Then, he turned to the other side, shifted to partner 2, and performed another roundhouse kick with the other lead leg. Thereafter, he returned to the center, moved to partner 3 in guard position, and performed a double roundhouse kick. Finally, the athlete ran backward to the start/finish line [22]. The performance time was measured with two sets of single-beam timing lights (Brower Timing Systems, Salt Lake City, UT, USA). Three trials were performed by each athlete, and the best performance was used for the analysis. The intra-class correlation coefficient (ICC) for test–retest trial for the present study was 0.85.

### 2.3. Statistical Analysis

Data were presented as mean and SD. The statistical analysis was performed using SPSS 20.0 statistical software (IBM corps., Armonk, NY, USA). The normality of data sets was checked and confirmed using the Kolmogorov–Smirnov test. Sphericity was tested and confirmed using the Mauchly test. Data were analyzed using a one-way analysis of variance with repeated measurements to compare performances throughout different experimental conditions. Bonferroni was used as post hoc test. Standardized effect size (Cohen’s *d*) analysis was used to interpret the magnitude of differences between variables and classified according to Hopkins [24]: *d* ≤ 0.2 (trivial), 0.2 < *d* ≤ 0.6 (small), 0.6 < *d* ≤ 1.2 (moderate), 1.2 < *d* ≤ 2.0 (large), 2.0 < *d* ≤ 4.0 (very large), and *d* > 4.0 (extremely large). The statistical significance level was set at *p* ≤ 0.05.

## 3. Results

Table 1 presents the performances recorded during the taekwondo-specific tests over different conditions.

There was no difference in the CMJ height between conditions (*F*_6182_ = 0.609 and *p* = 0.723). However, a significant difference in the number of kicks performed during the FSKT-10s was found (*F*_6182_ = 4.781 and *p* < 0.001) with the plyometric condition using 1:7 and SSRT ratios, which induced increased performances compared with the control condition (95%CI_dif_: 0.73; 4.23 and 0.73; 4.23, *d* = 1.15 and 1.14 (moderate), both *p* < 0.001, respectively). Similarly, agility performance differed across conditions (*F*_6182_ = 3.83 and *p* = 0.001) with the plyometric condition using 1:7 and SSRT ratios inducing improved agility (i.e., shorter time to complete the test) compared with the control condition (95%CI_dif_: −1.39; −0.22 and −1.34; −0.17, *d* = −0.97 and −1 (moderate), both *p* < 0.01, respectively). Concerning the total number of kicks recorded during the FSKT-mult, there was a condition effect (*F*_6182_ = 6.107 and *p* < 0.001) with lower values in the control condition compared with RHIT1:6 (95%CI_dif_: −13; −2, *d* = −1.42 (large) and *p* = 0.001), RHIT1:7 (95%CI_dif_: −13; −2, *d* = −1.42 (large) and *p* < 0.001), RHITSSRT (95%CI_dif_: −12; −1, *d* = −1.24 (large) and *p <* 0.05), P1:7 (95%CI_dif_: −14; −3, *d* = −1.21 (large) and *p* < 0.001) and PSSRT conditions (95%CI_dif_: −14; −3, *d* = −1.4 (large) and *p* < 0.001). Moreover, regarding the DI, there was a condition effect (*F*_6182_ = 3.079 and *p* = 0.007) with lower DI values recorded in the RHIT1:6 condition compared with the control (95%CI_dif_: −6.04; −0.49, *p* < 0.01 and *d* = −1.16 (moderate)), RHIT1:7 (95%CI_dif_: −5.69; −0.13, *p* < 0.05 and *d* = −1.03 (moderate)), P1:6 ratio (95%CI_dif_: −5.91; −0.35, *p* < 0.05 and *d* = −1.07 (moderate)), and PSSRT conditions (95%CI_dif_: −5.61; −0.057, *p* < 0.05 and *d* = −1.03 (moderate)).

## 4. Discussion

The present study investigated the effects on taekwondo athletes’ performance associated with different conditioning activities (i.e., plyometric and repeated high-intensity technique) using different work-to-rest ratios during such activities (1:6, 1:7 and SSRT). The main findings of the present study were that conditioning activities did not affect muscle power, but plyometrics using 1:7 work-to-rest ratio resulted in a higher number of kicks executed in the FSKT-10s and increased agility in the TSAT compared with the control condition. Moreover, except for the P1:6 condition, the total number of kicks during the FSKT-mult was higher after all experimental conditions in comparison with the control condition. Furthermore, decrement index in the FSKT-mult was lower after RHIT1:6 compared with the control, RHIT1:7, P1:6, and P (SSRT) conditions.

The absence of performance change in the CMJ observed in the present study is similar to that reported by other investigations [13,15], which showed that jump performance was not improved following plyometric, strength, or combined exercise-based conditioning activities in taekwondo athletes [13,15]. This result can be attributed to the lack of specificity of CMJ as well as its short duration [13,15]. Indeed, it was suggested that taekwondo athletes submitted to PAPE conditions would improve their performance only in specific tasks or actions lasting approximately 10 s [15]. However, other studies on combat sports [10,11] reported results contrary to those found in the present study. Specifically, Margaritopoulos et al. [10] showed that three sets of five tuck jumps induced a 3.5% significant improvement in CMJ height compared with the control condition after 5 min of the intervention in elite karate athletes. Moreover, Cimadoro et al. [11] showed that repeating 20 kicks every 3 s as hard as possible in Muay Thai athletes by alternating the right and left lower limb induced a significant increase (3.3 ± 3.0%) in CMJ height only after 5 min of rest interval. According to the findings from the meta-analysis of Wilson et al. [7], the volume of the conditioning activity and the recovery durations between this activity and the subsequent performance could be considered as determinants of performance achievement. Thus, the difference between our results and those ones [10,11] could be related to the duration of rest between the conditioning activity and the subsequent test as well as to the duration of the applied effort (3 s in the study by Cimadoro et al. [11] vs. 5 s in our study).

The 10 s frequency of the kick test is a practical tool that mimics many characteristics of taekwondo combat (e.g., body position and the use of the most applied kick) and has been used to assess the anaerobic performance of taekwondo athletes [14,15]. In the present study, three sets of consecutive vertical jumps over 40 cm obstacles over 5 s using 1:7 and SSRT ratios resulted in a higher number of kicks during FSKT-10s compared with the control condition. Similar results have been reported in the literature [8,9] using plyometric-based conditioning activity. In the present study, the high number of kicks recorded for the two plyometric conditions with a ratio of 1:7 and SSRT could be likely attributed to the increase in muscle temperature, which reduces the viscous resistance of muscles and joints and increases nerve conduction speed [4]. Moreover, PAPE is associated with the increase in the intracellular water, which can enhance muscle force production, especially in type II fibers [4]. However, Da Silva Santos et al. [15] showed that 3 sets of 10 vertical jumps over a 40 cm obstacle did not induce significant improvements in the FSKT-10s test compared with the control condition 10 min after the intervention in expert taekwondo athletes. In that study, only a combination of strength and plyometric exercises (i.e., a complex method) with 10 min intervals resulted in improved performance compared with the control, strength-only with self-selected rest, and plyometric-only with 5 min interval. The discrepancy between our findings and those of Da Silva Santos et al. [15] could be related to inter-individual variations or a difference in the motivation to exercise between participants.

For taekwondo athletes, the FSKT-mult test seems to be a good tool to discriminate performance [14], as taekwondo athletes who perform the best in such a test are more likely to perform high-intensity movements throughout the taekwondo match and increase their own chances of winning [23]. In the present study, except for the P1:6 ratio conditions, the total number of kicks during the FSKT-mult was higher after all other experimental conditions compared with the control condition. Furthermore, DI obtained from the FSKT-mult was significantly lower following the RHIT1:6 compared with the other conditions. The enhancement in performance during the FSKT-mult could be related to the fact that the practice of one task (e.g., conditioning contractions) can transfer to another similar task (i.e., skill transfer; [25]). It is known that balance between rest and effort is needed to generate potentiation [7]. Furthermore, it was considered that the availability of adenosine triphosphate and phosphocreatine energy reserves and the ability to perform successive high-intensity actions are key elements to achieve good performance in taekwondo [1]. Based on these observations, the alternation between effort and recovery periods during RHIT-based conditioning activity with a 1:6 ratio was favorable to ensure the availability of energy reserves, mainly from the phosphagen system. Therefore, the 1:6 ratio was the most favorable for maintaining an optimal balance between PAPE and fatigue or motor pattern interference (perseveration) effects. This explains the reduction in DI compared to plyometric and RHIT conditions with 1:7 and SSRT ratios. Our results are not in agreement with those reported by Da Silva Santos et al. [14], who showed that using strength exercise with different volumes and intensities had no effect on the total number of kicks nor on the DI, which varied from 13% to 19%. This difference in conditioning activity nature (i.e., specific technique vs. strength exercise) can explain the difference between our results and those reported by that study. Regarding the plyometric-based conditioning activity, the results of the present study are in agreement with those reported by Castro-Garrido et al. [13], which showed that neither the total number of kicks nor the fatigue index was significantly changed after 3 sets of 10 jumps over obstacles 20 cm high.

To improve agility, PAP has been proposed as a relevant strategy in current scientific research [20]. Due to the lack of similar protocols in combat sports that investigated the effect of such conditioning activity on athletes’ agility, it was difficult to compare our results with others. Furthermore, it is well known that plyometric action involves faster shortening–stretching cycles and increases movement speed [9]. With respect to voluntary muscle contractions, PAPE might result from increases in spinal-level excitability motoneuron output [4]. From these perspectives, the specific agility performance enhancement after plyometric conditions with 1:7 and SSRT ratios was consistent with the theory of neural enhancements resulting in increased recruitment of higher order motor units [26], better synchronization of the motor units involved, and reduction in presynaptic inhibition [6], which could all contribute to faster changes in direction. Finally, we acknowledge a limitation in the present study: the choice of a single rest interval (10 min) between the conditioning activity and the tests can be considered as a factor that can affect the results.

## 5. Conclusions

The current study showed that the inclusion of repeated high-intensity techniques and vertical jumps using taekwondo-specific work-to-rest ratios in warm-up sessions were effective to reduce the kick decrement index, improve taekwondo-specific agility, and increase the number of techniques performed by taekwondo athletes in a 10 s specific test. Specifically, (1) plyometrics with 1:7 work-to-rest ratio or self-selected rest time can improve taekwondo-specific agility and kicks-number, (2) repeated high-intensity techniques with a 1:6 ratio are beneficial to maintaining a low kicking-decrement index, and (3) plyometric and repeated techniques with different work-to-rest ratios enhance kicks-number. These exercises can be adopted by coaches as a specific warm-up strategy before competitions because of their benefits as well as their low logistical demands. Coaches can also include these exercises in athletes’ daily warm-up regime to increase performance output during training sessions. Finally, it is also possible for coaches to modify the work-to-rest ratios and the rest interval between these activities and subsequent performance depending on the conditioning activity used in order to prepare athletes to manage the constraints of the competition. Based on the present study’s results, coaches and sport scientists in combat sports and other types of sports can use specific drills as conditioning activities during warm-up to induce post-activation performance enhancement. Moreover, organizing these activities by applying specific work-to-rest ratios seems to be a good strategy that should be investigated further.

## Figures and Tables

**Figure 1 ijerph-19-01764-f001:**
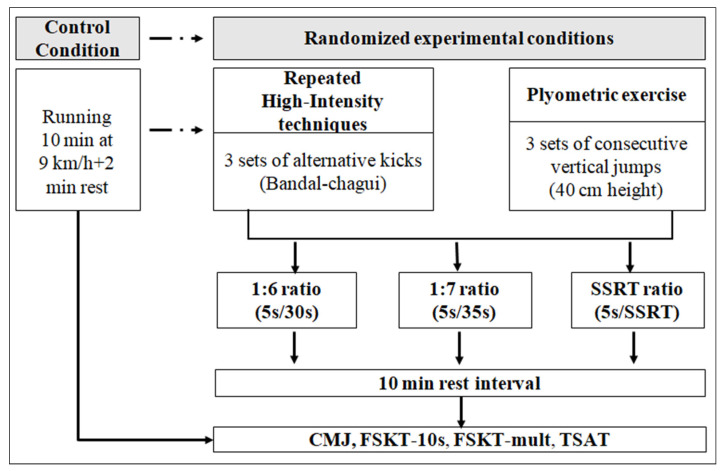
Schematic representation of the study design. SSRT: self-selected rest time, CMJ: countermovement jump test, CA: conditioning activity, TSAT: taekwondo-specific agility test, FSKT-10s: 10 s frequency speed of kick test, FSKT-mult: multiple frequency speed of kick test.

**Table 1 ijerph-19-01764-t001:** Countermovement jump (CMJ), 10 s frequency speed of kick test (FSKT-10s) repetitions, decrement index (DI), and the total number of kicks during multiple sets of FSKT (FSKT-mult) and taekwondo specific agility (TSAT) test during different conditions (values are mean ± SD; n = 27).

		CMJ (cm)	FSKT-10s (n)	FSKT-Mult		TSAT (s)
				DI (%)	Total Number of Kicks (n)	
	**Control condition**					
		25.9 ± 7.7	20 ± 2	8.9 ± 3.5	95 ± 6	7.0 ± 0.9
	**Repeated high-intensity technique**					
**1:6**		26.6 ± 7.4	21 ± 2	5.6 ± 2.2 ^a,b,c,d^	102 ± 5 ^¥^	6.6 ± 0.7
**1:7**		26.6 ± 7.8	20 ± 2	8.5 ± 3.5	103 ± 5 ^†^	6.6 ± 0.7
**SSRT**		27.0 ± 8.1	21 ± 2	8.0 ± 3.6	101 ± 5 ^§^	6.5 ± 0.7
	**Plyometric exercise**					
**1:6**		24.2 ± 8.2	21 ± 2	8.7 ± 3.7	100 ± 8	6.6 ± 0.6
**1:7**		27.6 ± 6.8	22 ± 2 ^†^	8.2 ± 3.2	103 ± 8 ^†^	6.3 ± 0.7 ^¶^
**SSRT**		25.6 ± 6.7	22 ± 2 ^†^	8.4 ± 3.3	104 ± 7 ^†^	6.3 ± 0.6 ^¶^

^†^ higher than the control condition (*p* < 0.001); ^§^ higher than the control condition (*p* < 0.05); ^¥^ higher than the control condition (*p* = 0.001); ^¶^: lower than the control condition (*p* < 0.01); ^a^ lower than the control condition (*p* = 0.008); ^b^ lower than repeated high-intensity technique using 1:7 ratio (*p* = 0.031); ^c^ lower than plyometric exercise using 1:6 ratio (*p* = 0.014); ^d^ lower than plyometric exercise using SSRT ratio (*p* = 0.041); SSRT: self-selected rest time.

## Data Availability

The data presented in this study are available on request from the corresponding author.

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
