# Peer review of "Acute Effects of Different Activity Types and Work-To-Rest Ratio on Post-Activation Performance Enhancement in Young Male and Female Taekwondo Athletes"

_ijerph, 2022, doi:10.3390/ijerph19031764_

Round 1
Reviewer 1 Report
The subject of this research work is an interesting and current topic. The article is largely written in a comprehensible manner, but a few points need to be noted that should still be considered.
No inferential statistical tests are given, Which procedure was used (MANOVA?) Please state this specifically.
Posthoc tests would also have to be reported.
How exactly was the comparison made within and between groups? The indication of the DF of the F-value indicates a 7-group comparison. These were both repeated measures (within) and between-group comparisons, weren't they?
Table 1 is very confusing. Certainly, a lot of information has to be contained in a small space, nevertheless, one has to search very hard for the exact effects.
While reading, it becomes partly unclear what the actual research question was. Is it now about the proof of optimal training interventions or is it about the validation of various test procedures. This should be explicitly clarified in prior and the discussion should then explicitly consider the findings in the light of the formulated research question.
Author Response
Response to Reviewer 1 Comments
Does the introduction provide sufficient background and include all relevant references?
(x) Can be improved
Please read below comments to specific points.
Is the research design appropriate?
(x) Can be improved
Please read below comments to specific points.
Are the methods adequately described?
(x) Must be improved
Please read below comments to specific points.
Are the results clearly presented?
(x) Must be improved
Please read below comments to specific points.
Are the conclusions supported by the results?
(x) Must be improved
Please read below comments to specific points.
Point 1: No inferential statistical tests are given. Which procedure was used (MANOVA?) Please state this specifically.
Posthoc tests would also have to be reported.
How exactly was the comparison made within and between groups? The indication of the DF of the F-value indicates a 7-group comparison. These were both repeated measures (within) and between-group comparisons weren't they?
Response 1: We thank expert reviewer for her/his suggestion. One-way analysis of variance with repeated measurements was used. Bonferroni was used as post-hoc test. Section 2.3. Statistical analysis was added.
Point 2: Table 1 is very confusing. Certainly, a lot of information has to be contained in a small space, nevertheless, one has to search very hard for the exact effects.
Response 2: Table 1 notes were changed to make effects more evident.
Point 3: While reading, it becomes partly unclear what the actual research question was. Is it now about the proof of optimal training interventions or is it about the validation of various test procedures? This should be explicitly clarified in prior and the discussion should then explicitly consider the findings in the light of the formulated research question.
Response 3: We thank expert reviewer for his suggestion. The actual research question was about the proof of optimal training interventions. This was further clarified in "1. Introduction" and considered in the light of the findings in "5. Conclusions".
We hope that the manuscript has now reached the standard necessary for formal acceptance endorsement in International Journal of Environmental Research and Public Health.
We look forward to hearing from you.
Best regards
Reviewer 2 Report
The manuscript was prepared very well. The introduction section justifies the purpose of the study. I congratulate the authors for the preparation of the manuscript
However, I have the following comments:
Introduction
- Line 76 add reference
- Explain: What do plyometrics consist of?
- It should include a hypothesis as to why this study is being conducted and how it differs from others that have been conducted.
Materials and Methods
The methodology is perfectly described and carried out
Results
- The tables/figures and the text describing them do not require any input, it is the strongest part of this study.
Discussion
- What specifically does this manuscript contribute?
- Include a limitations section.
- From the exercise described, what application would it have in other types of sport?
-Can it be applied to predict sport performance?
Author Response
Response to Reviewer 2 Comments
Does the introduction provide sufficient background and include all relevant references?
(x) Can be improved
Please read below comments to specific points.
Are the conclusions supported by the results?
(x) Can be improved
Please read below comments to specific points.
Point 1: Line 76 add reference.
Response 1: We thank the expert reviewer for her/his suggestion. Reference was added.
Point 2: Explain: What do plyometrics consist of?
Response 2: Definition of plyometrics was added.
Point 3: It (1. Introduction) should include a hypothesis as to why this study is being conducted and how it differs from others that have been conducted.
Response 3: 1. Introduction aim paragraph was changed as follows:
“Both taekwondo professionals and scientists aim to prepare athletes to better handle the competition demands. To do this, manipulation of different conditioning activities and work-to-rest ratios can be pursued to administer more effective training. To the current authors’ knowledge, no studies have investigated the combined effects of using plyometrics (P) or repeated high intensity techniques (RHIT) with different work-to-rest ratios during warm-up sessions on subsequent specific taekwondo performances. For this reason, the present study aims at comparing the effects of six different conditioning activities (P vs. RHIT) using different work-to-rest ratios (1:6, 1:7 and self-selected rest time, SSRT) and a control condition on physical performances in young taekwondo athletes. Due to the beneficial effects of the specific technical work on the technical performance of taekwondo athletes [16], it was hypothesized that RHIT with a specific ratio (1:6 or 1:7) would potentiate performance in 10-s frequency of kicks test (FSKT-10s) and in its multiple version (5x10s with 10-s intervals, FSKT-mult). However, as plyometric exercise was shown to be more appropriate to improve jump ability [10] and agility performance [20], it was hypothesized that plyometric-based conditioning activity would result in better jump height and agility performances with all used ratios.”
Point 4: What specifically does this manuscript contribute?
Response 4: Specific findings were highlighted in "5. Conclusions" as follows:
“Specifically, 1) plyometrics with 1:7 work-to-rest ratio or self-selected rest time can improve taekwondo-specific agility and kicks-number, whereas 2) repeated high-intensity techniques with 1:6 ratio is beneficial to maintain low kicking decrement index and 3) plyometric and repeated-techniques with different work-to-rest ratios enhance kicks-number.”
Point 5: Include a limitations section.
Response 5: A limitation is highlighted at the end of the discussion section as follows:
.“Finally, we acknowledge a limitation in the present study. In fact, the choice of a single rest interval (10 min) between the conditioning activity and the tests can be considered as a factor that can affect the results.”
Point 6: From the exercise described, what application would it have in other types of sport?
Can it be applied to predict sport performance?
Response 6: We thank expert reviewer for his suggestions. Following sentences were added to 5. Conclusions:
“Based on the present study’s results, coaches and sports scientists in combat sports and other types of sport can use specific drills as conditioning activities during warm-up to induce post-activation performance enhancement. Moreover, organizing these activities by applying specific work-to-rest ratios seems to be a good strategy that should be investigated further.”
We hope that the manuscript has now reached the standard necessary for formal acceptance endorsement in International Journal of Environmental Research and Public Health.
We look forward to hearing from you.
Best regards
Reviewer 3 Report
Generally, the manuscript is written very well, but I have some method recommendations.
Information about allocation concealment is missing.
The source of participants is needed (highlight them).
Inclusion and exclusion criteria are needed
Information about blinding is needed.
Franchini has a lot of autocitations
Add a limitation in discussion
Author Response
Response to Reviewer 3 Comments
Is the research design appropriate?
(x) Must be improved
Please read below comments to specific points.
Are the methods adequately described?
(x) Must be improved
Please read below comments to specific points.
Point 1: Information about allocation concealment is missing.
Information about blinding is needed.
Response 1: We thank the expert reviewer for her/his suggestion. The following sentences were added in the text:
“Due to the study design, allocation concealment as well as blinding would not be executed, as the athletes were aware of the condition being performed.”
Point 2: The source of participants is needed (highlight them).
Response 2: A convenience sampling was used with recruited participants belonging to the same club. Information was added in the text.
Point 3: Inclusion and exclusion criteria are needed.
Response 3: The following sentences were added in the text:
“To participate in the study, all athletes had to meet the following inclusion criteria: (i) should be young taekwondo athletes; (ii) should have five years or more of taekwondo experience; (iii) present no history of disease, medication and no lower limb injuries or neuromuscular disorder; (iv) athletes should not be in a period of body mass reduction process; and (v) females could participate if they were in the follicular phase of their menstrual cycle during the experimentation.”
Point 4: Franchini has a lot of autocitations.
Response 4: We kindly disagree, because these references were only used as they represent relevant studies (that are few and may be limited to those already cited in the present study) related to the topic and the combat sport investigated in the present study.
Point 5: Add a limitation in discussion.
Response 5: We thank expert reviewer for his/her suggestion. A limitation of the study is in the discussion and highlighted as follows:
“Finally, we acknowledge a limitation in the present study. In fact, the choice of a single rest interval (10 min) between the conditioning activity and the tests can be considered as a factor that can affect the results.”
We hope that the manuscript has now reached the standard necessary for formal acceptance endorsement in International Journal of Environmental Research and Public Health.
We look forward to hearing from you.
Best regards
Round 2
Reviewer 1 Report
The manuscript was significantly improved. The suggestions were well implemented. In my opinion the paper is now suitable for publication.